# Embryonic Nicotine Exposure Disrupts Adult Social Behavior and Craniofacial Development in Zebrafish

**DOI:** 10.3390/toxics10100612

**Published:** 2022-10-15

**Authors:** Gissela Borrego-Soto, Johann K. Eberhart

**Affiliations:** Department of Molecular Biosciences, School of Natural Sciences, University of Texas at Austin, Austin, TX 78713, USA

**Keywords:** nicotine, embryonic exposure, adult zebrafish, social behavior, craniofacial defects, cotinine

## Abstract

Cigarette smoking remains the leading cause of preventable death and morbidity worldwide. Smoking during pregnancy is associated with numerous adverse birth outcomes, including craniofacial and behavioral abnormalities. Although tobacco smoke contains more than 4000 toxic substances, nicotine is addictive and is likely the most teratogenic substance in cigarette smoke. However, much remains to be determined about the effects of embryonic nicotine exposure on behavior and craniofacial development. Therefore, this study evaluated adult social behavior in zebrafish, craniofacial defects, and nicotine metabolism in embryos after embryonic nicotine exposure. Zebrafish embryos were exposed to different doses of nicotine beginning at 6 h post fertilization. To evaluate craniofacial defects, the embryos were collected at 4 days post fertilization and stained with Alizarin Red and Alcian Blue. For behavioral testing, embryos were reared to adulthood. To evaluate nicotine metabolism, cotinine levels were analyzed at various time points. Our findings demonstrate that embryonic exposure to nicotine modifies social behavior in adulthood, causes craniofacial defects with reduced size of craniofacial cartilages, and that zebrafish metabolize nicotine to cotinine, as in humans. Together, our data suggest that zebrafish are useful as a model for studying nicotine-related diseases.

## 1. Introduction

The tobacco epidemic is one of the biggest public health problems worldwide, killing more than 8 million people per year, including around 1.2 million deaths from exposure to second-hand smoke [1]. Despite the implementation of tobacco control policies, approximately 34 million people still currently smoke cigarettes. More than half of smokers make a serious attempt to quit once a year; however, only 3–5% maintain abstinence after one year [2]. Tobacco use usually begins during adolescence, and one-third of those adolescents become nicotine dependent [3]. Most women who started smoking during adolescence, continue smoking during pregnancy and find it hard to quit. Although the consumption of cigarettes has decreased in recent years, the use of electronic cigarettes has increased. Women often perceive electronic cigarettes as a safe alternative to conventional cigarettes, and switch to e-cigarettes in pregnancy as a means of smoking cessation. However, e-cigarettes still contain nicotine. Nicotine is the addictive substance in tobacco which readily crosses the placenta and blood-brain barrier. Nicotine is considered the principal component of tobacco responsible for the adverse effects on the developing fetus. Like acetylcholine, nicotine binds to nicotinic acetylcholine receptors in the brain to activate the release of neurotransmitters such as dopamine, norepinephrine, serotonin, glutamate, and GABA [4], but the effects and mechanisms of nicotine on the developing embryo are not clear.

Smoking during pregnancy is linked increased risk of adverse birth outcomes, including low birth weight, prematurity, neonatal mortality, and abnormal childhood behavior [5]. Among the most common craniofacial defects associated with smoking during pregnancy is non-syndromic cleft lip with or without cleft palate, with an odds ratio of 1.22 [6] and 1.5 for passive smoking [7]. Prenatal tobacco exposure has been associated with hyperactivity and negative and externalizing behaviors in children, such as aggression, overactivity, and reduced social behavior [8,9]. It also associates with a higher probability of experimenting with drugs during adolescence [10], and higher rates of delinquency and criminal behavior [11]. It is important to understand the effect of prenatal nicotine exposure and how it disrupts the genesis of birth defects and behavior. This understanding requires faithful animal models.

Zebrafish are recognized as useful models for neurodevelopmental and craniofacial disorders, toxicology, and behavior. Zebrafish embryos have been used to study the effects of various chemical exposures on craniofacial development [12,13,14,15]. Furthermore, zebrafish are highly social, and so are useful for studying how chemical exposures can alter behavior. For instance, embryonic exposure to ethanol has been shown to disrupt adult social behavior [16,17] across zebrafish strains [18]. However, little is known about the effects of embryonic nicotine exposure and developmental effects in zebrafish.

Here, we examine nicotine oxidation in zebrafish embryos, and the effect of embryonic nicotine exposure on craniofacial and behavioral development in zebrafish. Our results demonstrate that the zebrafish embryo is able to metabolize nicotine. We show that embryonic nicotine exposure disrupts craniofacial development and causes social deficits and hyperactivity in adults, making zebrafish useful for the study of nicotine-related developmental defects.

## 2. Materials and Methods

### 2.1. Chemicals

Nicotine (Nicotine hydrogen tartrate salt, purity, ≥98%, CAS number 65-31-6, Sigma-Aldrich, Burlington, MA, USA). Stock solutions of 100 mM nicotine were prepared in DMSO (Dimethyl Sulfoxide, purity, ≥99.9%, CAS number 67-68-5, Fisher Scientific, Waltham, MA, USA) and stored at −20 °C until use.

### 2.2. Zebrafish Husbandry

This study was performed in accordance with recommendations in *The Zebrafish Book*, 5th edition [19], and the *Guide for the Care and Use of Laboratory Animals of the National Institutes of Health*. The protocol was approved by the University of Texas at Austin Institutional Animal Care and Use Committee. All zebrafish were housed in a recirculating aquatics facility in 3 L tanks at a density of 8–15 fish at the University of Texas, Austin under IACUC-approved conditions. All zebrafish received daily health checks and we observed no sickly fish in any groups.

Wild-type zebrafish derived from the AB strain were maintained at 28 °C under a 14:10 light/dark period. Embryos were maintained in embryo media (EM) in a 28 °C incubator. Zebrafish were staged using the published staging guide for zebrafish [20].

### 2.3. Embryo Exposure and Experimental Design

Nicotine solutions at 12.5 µM, 25 µM, and 50 µM were prepared by diluting nicotine stock in EM. Zebrafish embryos were exposed to doses of nicotine beginning at 6 h post fertilization (hpf). The control group was treated with 0.05% DMSO in EM. The embryos were kept in a 28 °C incubator in 22 mm petri dishes and were cleaned daily. For craniofacial analyses, fish were removed from the nicotine solution at 4 dpf and processed for skeletal staining. For behavioral testing, zebrafish were removed from their solution at 5 dpf, reared to adulthood, and tested between 5–6 months old.

### 2.4. Cotinine Assay

To evaluate if zebrafish embryos metabolize nicotine similarly to humans, cotinine concentration in embryonic lysates was analyzed. Cotinine levels were analyzed following an exposure starting at 4 dpf with different nicotine concentrations (12.5, 25, and 50 µM). Embryos were collected at 6, 12, and 24 h following nicotine exposure. For analysis of early nicotine metabolism, embryos were exposed to 50 µM nicotine at 6 hpf and collected at 24, 25, 26, 36, 48, and 72 hpf. Embryos were lysed by incubating the samples in 25 µL of lysis buffer (1 M Tris pH = 8.3, 1 M KCl, 1 M MgCl2, 20% Tween-20, 10% NP40) for 3 h at 55 °C followed by 20 min at 94 °C. The lysates were stored at −20 °C until use. Cotinine quantification was performed using a Cotinine ELISA kit (Calbiotech, Inc., Spring Valley, CA, USA) following manufacturer protocols. The absorbance was read on a Synergy H4 Hybrid Multi-Mode Microplate Reader. Three biological replicates were collected from each sample and were measured in triplicate.

### 2.5. Dual Bone and Cartilage Staining

Cartilages and bones were stained using a previously described Alizarin Red and Alcian Blue staining protocol [21]. Briefly, embryos were fixed for 1 h in 2% paraformaldehyde in phosphate-buffered saline, followed by dehydration into 100% ethanol and overnight staining with Alcian Blue. Next, embryos were rehydrated and bleached with 3% H_2_O_2_/0.5% KOH and then stained with Alizarin Red. Finally, embryos were cleared and stored in 50% glycerol and 0.2% KOH. Imaging was performed on a Zeiss Axio Imager. Whole-mount and flat-mount images were collected. Neurocranial and viscerocranial cartilages were measured using ImageJ software for Windows version 1.8.0.

### 2.6. Behavioral Assay

Adult social behavior analyses were performed according to a well-established protocol generated by Fernandes et al. [17]. Briefly, testing tanks were 37 L (50 × 25 × 30 cm, L × W × H) with the back and bottom coated inside with white corrugated plastic to avoid reflections. Aquaneering tanks of 1.4 L (ZT140) were placed on both sides along the width of each test tank. One of the tanks contained only water, and another contained 2 male and 2 female fish, serving as the stimulus tank. The outer walls of the ZT140 tanks were also covered with corrugated plastic. Opaque white barriers were placed between the test tanks and the ZT140 tanks to visually isolate the experimental fish during the habituation period. The behavior of experimental fish was recorded and analyzed in real time using Ethovision. The test tank was divided into 10 virtual zones of 10 cm each in Ethovision software to analyze the time that the fish spent in each zone during habituation and the stimulus period. Zone 1 was defined as the zone closest to the stimulus. Half of the fish in each treatment were tested with the stimulus on the right side, and the other half on the left side, to eliminate the possibility of a side bias.

For each trial, an experimental fish was placed into the test tank and 30 s later a 20-min recording session was started. The first 10 min of recording consisted of a habituation period, at the end of 10 min, barriers between the test tanks and the ZT140 tanks were removed, and in the following 10 min social behavior was quantified.

### 2.7. Data Analysis

One-way ANOVA with Tukey’s multiple comparisons correction was performed for determination of significance in experimental groups for behavioral assays and craniofacial measurements. An unpaired, two-tailed t-test was used to calculate significance between two groups. Two-way ANOVA with Tukey’s multiple comparisons correction was performed to test for differences in behavior between males and females. Data are presented as means ± SEM; *p* < 0.05 was considered statistically significant. Graphpad Prism 9.4.0 (San Diego, CA, USA) was used for statistical analysis and graphing.

No statistical differences were observed between males and females in any of the variables analyzed during either the habituation or stimulus period (Appendix A). Therefore, all data contain both sexes.

## 3. Results

### 3.1. Nicotine Is Metabolized by Zebafish Embryo

In humans, approximately 75% of nicotine is converted to cotinine [22]. In order to determine if zebrafish metabolize nicotine similarly to humans, we examined cotinine levels in nicotine-exposed zebrafish embryos. Nicotine is metabolized mainly in the liver in humans, and liver growth and development is extensive between 3 dpf and 5 dpf in zebrafish [23]. Therefore, we exposed 4 dpf larvae to different nicotine concentrations, and assessed cotinine production at 6, 12, and 24 h post exposure. Five embryos were used to measure cotinine via ELISA at each time point. Cotinine concentration increased in a time- and dose-dependent fashion (Figure 1A). At 24 h, after 50 µM nicotine exposure, cotinine concentration increased up to 46.8 ng/mL (SEM ± 1.5) in 5 dpf fish.

To determine when zebrafish embryos first metabolize nicotine, we exposed zebrafish embryos to 50 µM nicotine, beginning at 6 hpf and collecting embryos at 24, 25, 26, 36, 48, and 72 hpf. Figure 1B shows that the cotinine concentration at 24, 25, and 26 hpf had a mean of 2.9 ng/mL (SEM ± 0.3). The detection limit of the cotinine kit provided is 5 ng/mL, and these values mirror those in the control, DMSO-treated embryos. However, cotinine concentrations were 5.9 (SEM ± 0.52), 16.2 (SEM ± 0.69), and 41.2 (SEM ± 0.57) ng/mL at 36, 48, and 72 hpf, respectively (Figure 1B). Therefore, cotinine levels were first detected at 36 hpf, and continued to increase over time. Collectively, we concluded that zebrafish embryos metabolize nicotine to cotinine, as in humans, detectable by ELISA by at least 36 hpf.

### 3.2. Embryonic Nicotine Exposure Disrupts Craniofacial Development

Smoking during pregnancy has been associated with craniofacial defects, especially orofacial clefts in humans. To analyze the craniofacial effects produced by embryonic exposure to nicotine, zebrafish embryos were exposed to different concentrations of nicotine from 6 hpf to 4 dpf. After nicotine exposure, dual Alizarin Red and Alcian Blue staining was performed. Images were collected and viscerocranium and neurocranium cartilages were measured. Whole-mount and flat-mount images showed that nicotine caused a dose-dependent reduction in the craniofacial skeleton (Figure 2).

To quantify the effect of nicotine on the craniofacial skeleton, we took linear measurements of several craniofacial elements. Nicotine-exposed embryos showed reduction in viscerocranial cartilages (*p* < 0.0001) in a dose-dependent manner. The embryonic lower jaw (Meckel’s cartilage) (Figure 3B) and cartilages derived from the second pharyngeal arch (the ceratohyal and hyosymplectic) were all significantly reduced, relative to the controls, following exposure to 25 µM or 50 µM nicotine (Figure 3C,E). The embryonic upper jaw (palatoquadrate) was significantly reduced in all treatment groups (Figure 3D). Therefore, embryonic nicotine exposure causes significant cartilage reduction, especially at high concentrations (25 and 50 µM). A full table of ANOVA results can be found in Appendix A.

While the viscerocranium consists entirely of neural crest-derived skeletal elements, only the anterior neurocranium is crest-derived, with the posterior region being predominantly derived from mesoderm [24]. Total neurocranium length was reduced due to nicotine exposure at 25 and 50 µM (Figure 4B, *p* = 0.0021). Interestingly, this effect is driven by the anterior neurocranium (Figure 4C, *p* = 0.0002) and not the posterior neurocranium (Figure 4D, *p* = 0.2443), although we do note a trend in the posterior neurocranium. When only the anterior neurocranium is considered, even the 12.5 µM nicotine group shows a significant reduction relative to the controls (Figure 4C). The anterior neurocranium is subdivided into the ethmoid plate at the anterior tip and the trabeculae, which connect to the posterior neurocranium. No significant differences were found in length of the trabecula (Figure 4E, *p* = 0.0508). However, the ethmoid plate was reduced in length and width after embryonic nicotine exposure (Figure 4F, *p* < 0.0001; and Figure 4G, *p* = 0.0002). Thus, embryonic exposure to nicotine disrupts the development of the neural crest-derived craniofacial skeleton in zebrafish.

Neural crest cells differentiate into a variety of cell types like chondrocytes, neurons, and melanocytes [25]. To determine if other neural crest derivatives were disrupted by nicotine, we quantified melanocytes at 36 hpf in the control group and 50 µM nicotine in the exposed group. We found a statistically significant reduction in melanocytes in the head of the treated group versus the control group (*p* = 0.0474; DMSO M = 8.6, SEM ± 0.53; 50 µM Nicotine M = 6.8, SEM ± 0.45) (Appendix A).

To determine if nicotine caused a general developmental delay, we quantified overall body length in exposed fish. We found no significant differences in body length at 4 dpf (DMSO M = 2239.33 µm, SEM ± 16.68; 12.5 µM Nicotine M = 2174.76 µm, SEM ± 25.72, *p* = 0.1855; 25 µM Nicotine M = 2265.66, SEM ± 17.40, *p* = 0.8355; 50 µM Nicotine M = 2161.30, SEM ± 23.32, *p* = 0.0684) (Appendix A). Thus, the effect of nicotine is not uniform and neural crest-derived cell populations are particularly sensitive to disruption.

### 3.3. Social Behavior Is Affected by Embryonic Nicotine Exposure

To investigate the effect of embryonic nicotine exposure on social behavior, embryos were exposed to nicotine at different concentrations from 6 hpf to 5 dpf. Embryos were reared until adults under standard conditions. More than 90% of the fish exposed to either DMSO or nicotine at 12.5 and 25 µM survived to adulthood. In the 50 µM nicotine group only 27% survived despite being healthy at 5 dpf. Thus, at this higher dose, embryonic nicotine exposure has a dramatic effect on larval and/or postlarval survival, potentially due to effects on the craniofacial skeleton.

Social behavior was assessed in all groups (DMSO n = 22; 12.5 µM nicotine n = 22; 25 µM nicotine n = 30; 50 µM nicotine n = 23). Figure 5 shows the average distance between the live shoal and the test fish across each of the 20 min of the behavioral assay (for clarity, each experimental group is shown separately in comparison to the control). During the first 10 min (habituation) fish swam back and forth in the test tank in all groups, with an average distance of 25.8 cm (SEM ± 0.83), 25.2 cm (SEM ± 0.69), 25.5 cm (SEM ± 0.53), and 25.2 cm (SEM ± 0.68), for DMSO, 12.5 µM nicotine, 25 µM nicotine, and 50 µM nicotine groups, respectively (*p* = 0.9820). When the barrier was removed at 10 min, the test fish across all groups approached the live shoal; however, the 50 µM nicotine-exposed group did not maintain proximity as observed in the control group (*p* = 0.0167). Average distance between the stimulus and the DMSO, 12.5 µM nicotine, 25 µM nicotine and 50 µM nicotine groups was 10.2 cm (SEM ± 0.75), 8.0 cm (SEM ± 0.54), 12.6 cm (SEM ± 0.66), and 15.8 cm (SEM ± 0.75), respectively. While there appeared to be a trend for the 12.5 µM nicotine group to shoal more robustly than the DMSO group, we only found statistical differences at minute 10 and 11, where the DMSO fish shoals better than the 12.5 µM fish (Appendix A).

Figure 6 shows the average time fish spent in each zone during habituation (Figure 6A) and when the live shoal was visible (Figure 6B) in the control and experimental groups. During habituation, fish spent more time in the zones nearest either end of the tank (zones 1, 2, 9, and 10; *p* < 0.0001). When the live shoal was visible, the distributions of the time in each zone were skewed to the side of the tank adjacent to the live shoal in all groups (zone 1; *p* < 0.0001), although the percentage of time was reduced in fish exposed to 25 and 50 µM nicotine.

To quantify the effect of nicotine on social behavior, we compared the average time spent in the zone closest to the stimulus across groups (zone 1). As shown above, all groups spent most of the time in zone 1 near the stimulus. The control and fish exposed to 12.5 µM nicotine spent 59% (SEM ± 7.8) and 69% (SEM ± 5.6) of the time in zone 1, respectively. However, fish exposed to 25 and 50 µM nicotine spent 47% (SEM ± 4.7) and 36% (SEM ± 4.6) of the time in zone 1, respectively. An ANOVA test showed significant differences in the time spent in zone 1 across groups (*p* = 0.0152). The reduction in the 25 µM exposure group did not reach significance. However, Tukey’s analysis demonstrated that the reduction the 50 µM exposure group was significant relative to fish in the DMSO and the 12.5 µM nicotine groups (*p* = 0.0479 and *p* = 0.0202, respectively) (Figure 7). Therefore, embryonic exposure to nicotine reduces the shoaling response of adult zebrafish.

To characterize the initial response to the stimulus, we examined the length of time the fish took to reach zone 1 following the shoal presentation. While the control fish immediately approach the stimulus, nicotine-exposed fish are slower. Fish exposed to 12.5 µM and 25 µM nicotine as embryos took significantly longer than the DMSO group to reach zone 1 (*p* = 0.0011, M = 15.8 s, SEM ± 2.85; M = 20.6 s, SEM ± 4.0, *p* < 0.0001, respectively). While there were no differences between the control and 50 µM nicotine groups (50 µM nicotine, M = 7.9 s, SEM ± 2.7; DMSO: M = 0.4 s, SEM ± 0.19; *p* = 0.2514), differences were found between the 25 and 50 µM nicotine exposed groups (*p* = 0.0142), showing that fish in the 25 µM nicotine group take longer to reach zone 1 than the 50 µM nicotine group. Thus, embryonic nicotine exposure reduced the immediate response to the shoal, but not in the highest nicotine exposure group (Figure 8; see also Figure 5).

One possibility for the reduced time to reach zone 1 is that the fish move slower. Thus, we quantified the percentage of time the fish were moving, the distance traveled, and speed during habituation (Figure 9). No statistical differences were observed in the percentage of time moving between the groups (DMSO: M = 81.4%, SEM ± 1.4; 12.5 µM nicotine: M = 84.3%, SEM ± 3.1; 25 µM nicotine: M = 85.3, SEM ± 2.9; 50 µM nicotine: M = 86.4, SEM ± 1.1) (Figure 9A). Interestingly, 25 µM and 50 µM nicotine exposed fish swam further relative to the controls (M = 464.9 cm, SEM ± 17.6, *p* = 0.0061; M = 451.2 cm, SEM ± 21.4, *p* = 0.0491; respectively) (Figure 9B). There was a non-significant increase in distance in the 12.5 µM nicotine exposed group relative to DMSO (M = 369.4 cm, SEM ± 13.9; M = 436.9 cm, SEM ± 23.0; respectively). Similarly, 25 µM and 50 µM nicotine exposed fish swam faster (M = 8.0 cm/s, SEM ± 0.29, *p* = 0.0133; M = 7.9 cm/s, SEM ± 0.35, *p* = 0.0353; respectively) than the control and 12.5 µM nicotine exposed groups (M = 6.5 cm/s, SEM ± 0.22; M = 7.4 cm/s SEM ± 0.39; respectively) (Figure 9C). Here, once again, the 12.5 µM nicotine group displayed an increased trend. Thus, embryonic nicotine exposure may cause some hyperactivity in adults.

## 4. Discussion

We used zebrafish to determine behavioral and craniofacial outcomes following embryonic nicotine exposure. Our observations revealed that embryonic nicotine exposure disrupts social behavior and craniofacial development in a dose-dependent manner. We also found a dose-dependent increase in motor behavior. We show that zebrafish metabolize nicotine similarly to humans. Thus, our results demonstrate that zebrafish are a useful model for nicotine-induced birth defects.

Maternal smoking during pregnancy has been associated with a variety of adverse mental health and behavioral outcomes in childhood. These include anxiety disorders (AD), depressive disorders, attention-deficit/hyperactivity disorder (ADHD), autism spectrum disorder (ASD), and learning impairments, and even occur in concentrations that do not cause visible growth retardation [26,27,28]. However, available evidence in epidemiological studies is mixed [29]. Nicotine is the addictive component of cigarettes and e-cigarettes, and is considered to be the main teratogenic substance in cigarette smoke [30,31,32]. Studies of nicotine exposure in rodents have yielded inconsistent results, possibly due to variability in experimental methodology, such as the nicotine timing of exposure, nicotine dose, and administration route [33,34,35,36]. Thus, additional models are needed to gain an understanding of the consequences of embryonic nicotine exposure.

### 4.1. Nicotine Metabolism in Zebrafish

To validate zebrafish as a study model for embryonic nicotine exposures, we first characterized the metabolism of nicotine in zebrafish embryo lysates. Cotinine is the main nicotine metabolite, and has been widely used as a cigarette smoke exposure biomarker in humans. Our results show that the zebrafish embryo is able to oxidize nicotine to cotinine beginning by at least 36 hpf. After this time point, cotinine levels begin to increase over time.

Cotinine can cross the blood-brain barrier in zebrafish, but is a weak agonist of nicotinic acetylcholine receptors [37,38]. The effects of the combination of nicotine and cotinine during development are unclear and little studied. Baldwin et al. reported that nicotine and cotinine do not produce synergistic effects [39]. Dawson et al. also studied the toxic effects of nicotine and cotinine in *Xenopus leavis* embryos. They found that the metabolic conversion of nicotine to cotinine increased the concentration of nicotine needed to induce teratogenesis [40]. On the other hand, Bastianini et al. demonstrated that early exposure to nicotine or cotinine (independently) produces long-lasting sleep alterations and downregulation of hippocampal corticosteroid receptors in adult mice [41]. Thus, these data suggest that, although cotinine has no or less effects than nicotine during early development, early exposure to cotinine could produce longitudinal effects on behavior in adult fish. In this study, cotinine levels were detected after 36 hpf and levels were increased over time, so the nicotine-cotinine interactions in zebrafish embryos could also have deleterious effects on adult zebrafish. Future experiments specifically exposing zebrafish to cotinine, or other nicotine metabolites, will help shed light on this question.

Cotinine levels have been reported in human placenta from women who reported smoking during pregnancy, and cotinine modifies the expression of xenobiotic-metabolizing enzymes in fetal tissues [42]. Similarly, downstream metabolites quantified in maternal and cord sera (cotinine, norcotinine, and 3-hydroxycotinine) were associated with DNA methylation sites located on the *MYO1G*, *AHRR*, and *GFI1* genes [43]. Our results demonstrate that zebrafish embryos oxidize nicotine to cotinine, but we do not yet know if other similar downstream metabolites were generated. Quantitative studies of nicotine metabolite production, as well as toxicogenomic studies in zebrafish, are necessary to be able to perform association studies of developmental defects.

### 4.2. Nicotine and Craniofacial Defects

Smoking during pregnancy is associated with adverse reproductive outcomes, including low birth weight and craniofacial defects. Among craniofacial defects, non-syndromic cleft lip and/or palate is the most common [7,44,45]. Most studies suggest that about 70% of nonsyndromic cases are associated with environmental interactions, and have been particularly linked with smoking and even passive smoking [44,46]. In mice, nicotine exposure affects mandibular development with an increase in proliferation and a decrease in apoptosis, and inhibits palatal fusion [47,48,49]. In rats nicotine induces delayed chondrogenesis [50].

In this study, a dose-dependent reduction in cartilage size was observed in zebrafish embryos exposed to nicotine, especially in the anterior region of both the viscerocranium and the neurocranium. A decrease in length and width of the ethmoid plate was observed, but no significant differences were observed in the posterior region of the neurocranium; although a trend was evident. During zebrafish embryogenesis, cranial neural crest cells (CNCC) migrate from the dorsal neural tube to form pharyngeal arches; a portion of these cells migrate to the frontonasal region and the rest migrate to the first arch. The palate begins its morphogenesis between 36 and 48 hpf in zebrafish, and CNCC will populate the midline of the ethmoid plate [51,52]. The fact that the zebrafish palate appears to be most sensitive to nicotine indicates that zebrafish will be useful in elucidating the association of nicotine and cleft palate in humans.

Our results demonstrate that other neural crest derivatives are also sensitive to disruption due to nicotine exposure. We observed a reduced number of melanocytes in the 50 µM nicotine treated group vs DMSO group. Similarly, Parker et al. reported varied pigmentation in zebrafish larvae 10 dpf exposed to 40 µM [53]. Analyses of additional neural crest-cell-derived tissues will elucidate the extent to which the neural crest is generally sensitive to nicotine exposure.

In addition, we found high mortality in zebrafish adults exposed to 50 µM nicotine. Although we did not analyze feeding behavior, it is possible that these craniofacial defects, especially the reduced jaw, could lead to feeding defects, resulting in the high mortality in the 50 µM group.

A previous study of nicotine teratogenesis in zebrafish showed that notochord length was reduced in larvae treated every day with nicotine at 20 µM, but not at 40 µM, and dorsal curvature of the body axis increased as nicotine concentrations increased [53]. In our study, we did not observe these effects. In fact, most of the larvae at 50 µM showed a slight ventral curvature to the body axis. Additionally, no statistical differences were found in body length. The cause of these differences is unclear, but could reflect the effects of strain differences.

We conclude that CNCC-derived structures are particularly sensitive to nicotine-induced defects, as is the case with other teratogens, such as alcohol. Future experiments are needed to determine if these defects are due to alterations to apoptosis, proliferation, and/or specification.

### 4.3. Embryonic Nicotine Exposure and Adult Zebrafish Behavior

Studies have shown that nicotine exposure in zebrafish embryos affects the embryonic spinal motor circuit [54,55], increases locomotor phenotypes in zebrafish larvae [56], has neurotoxic effects, and decreases survival rates in zebrafish larvae [57]. Studies in adult zebrafish have shown that nicotine reduces anxiety-like behavior, when the adult fish is exposed to 0.3 mg/L for 20 min. Concentrations of cotinine 300-fold higher are necessary to have the same effect [37]. In contrast, chronic 4-day nicotine exposure (1–2 mg/L) of adult zebrafish induced robust anxiogenic behavioral responses and mildly increased adult zebrafish shoaling [58]. Hawkey et al. concluded that nicotine induces anxiolytic-like effects in adult zebrafish after acute (0.3–30 µM nicotine for 20 min) and subchronic treatment (0.3–30 µM nicotine 20 min per day for 5 days). Again, much higher cotinine doses are necessary to cause anxiolytic-like effects [59]. It is currently unclear if similar relationships between nicotine and its metabolites will occur from embryonic exposures. Future experiments are required to test this possibility.

The present study is the first behavioral analysis of adult zebrafish exposed to nicotine at embryonic stages. Our results show that embryonic nicotine exposure lessens social behavior in adults. This reduction is statistically significant at 50 µM, and we note that there is a trend at 25 µM. While there was high mortality in the 50 µM nicotine group, it is unlikely that this effect is due to the fish being generally sick. First, there was not elevated mortality in the 25 µM group. Second, fish in the 50 µM group spend an equal amount of time moving, as do the controls. The clear dose-dependent effects of nicotine on social behavior will facilitate identification of genetic and environmental modifiers to the effect of nicotine.

We also found differences in the initial approach to the shoal (latency) between the study groups. Embryos exposed to 12.5 and 25 µM nicotine took a longer time to reach the zone closest to the shoal than DMSO group. Interestingly, no differences were found between the 50 µM nicotine and DMSO groups; again suggesting that this was not a sickly group of fish. The alterations in latency are not due to impaired motor function because these fish actually move as fast (12.5 µM) or faster (25 and 50 µM) than the control fish, and may suggest ADHD-like phenotypes in these fish. It is not clear why the effect on latency disappears with an exposure to 50 µM nicotine. Though it is possible that mortality in this group selectively eliminated those fish that would have displayed this defect. Larval social behavior assays have been developed and could be used to test this possibility [15].

The cause of these nicotine-induced behavioral alterations remains to be determined. The increased latency to approach the shoal could suggest either motivational or sensory deficits were present. Visual assays, such as the optomotor response, along with analyses of retinas from these fish, would be useful to determine if there is sensory involvement. The shoaling deficit is reminiscent of that caused by embryonic alcohol exposure [60]. Alcohol and nicotine co-exposures are common. It will be of interest to determine if these two substances interact on the development of social behavior.

## 5. Conclusions

Our findings demonstrate that embryonic exposure to nicotine modifies adult social behavior, impacts hyperactivity, and disrupts craniofacial development. We find that zebrafish oxidize nicotine to cotinine, as in humans. Therefore, our findings demonstrate that zebrafish are a useful model for the study of nicotine-related diseases.

## Figures and Tables

**Figure 1 toxics-10-00612-f001:**
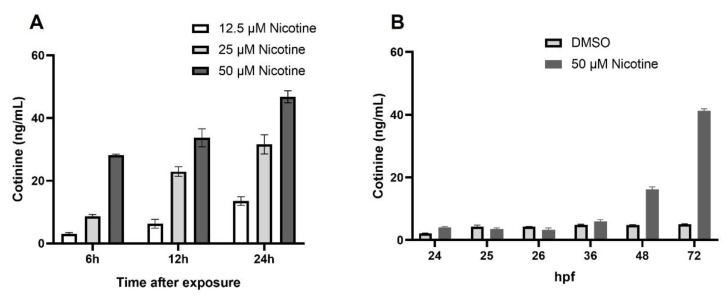
Cotinine levels in zebrafish embryo increases with the time of exposure. (**A**) Cotinine levels in zebrafish larvae exposed to different nicotine concentrations beginning at 4 dpf. (**B**) Cotinine levels in embryos exposed to 50 µM nicotine from 6 to 72 hpf.

**Figure 2 toxics-10-00612-f002:**
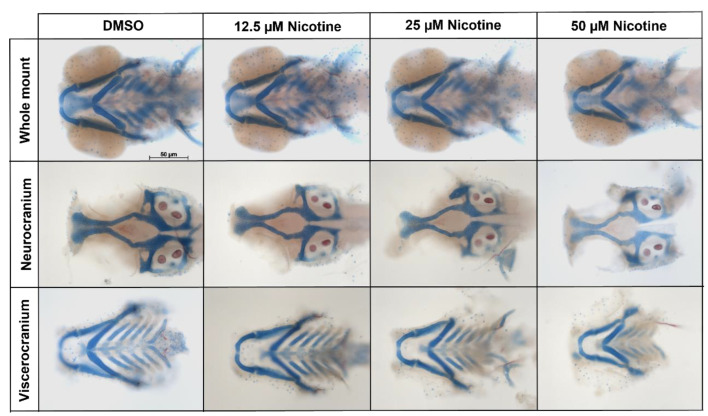
Embryonic nicotine exposure disrupts craniofacial development. Representative whole mount, neurocranial, and viscerocranial flat mount preparation after nicotine exposure from 6 hpf to 4 dpf. Decreases in the size of the craniofacial skeleton were observed as nicotine concentration increased. All images were captured at 10× magnification. Scale bar 50 µm.

**Figure 3 toxics-10-00612-f003:**
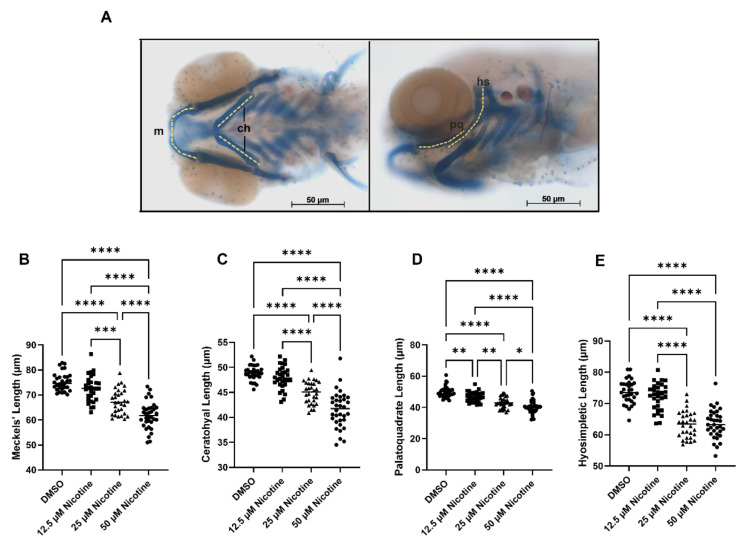
Embryonic nicotine exposure effects viscerocranial cartilages length. (**A**) Viscerocranial cartilages measurements. (**B**) Meckel’s length, (**C**) Ceratohyal length, (**D**) Palatoquadrate length, and (**E**) Hyosimpletic length. The size of all viscerocranial cartilages decreased with embryonic exposure to nicotine in a dose-dependent manner. Sample sizes are: DMSO n = 31; 12.5 µM nicotine n = 30; 25 µM nicotine n = 30; and 50 µM nicotine n = 34; m, meckel’s cartilage; ch, ceratohyal; pq, palatoquadrate; hs, hyosimpletic. * = *p* < 0.05; ** = *p* < 0.01; *** = *p* < 0.001; **** = *p* < 0.0001. All images were captured at 10× magnification. Scale bar 50 µm.

**Figure 4 toxics-10-00612-f004:**
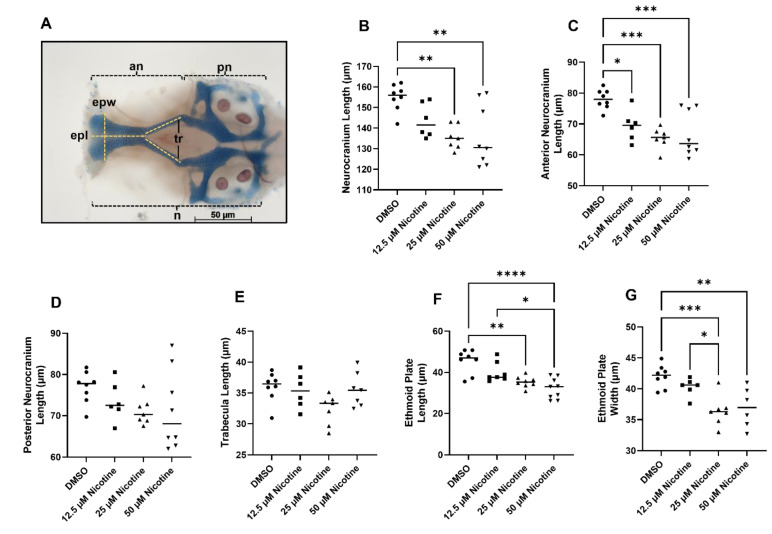
Embryonic nicotine exposure reduces cartilage in the neurocranium. (**A**) Neurocranium cartilages measurements. (**B**) Neurocranium total length was reduced following exposure to 25 or 50 µM nicotine. (**C**) The anterior neurocranium was reduced in all nicotine-exposed groups. (**D**) Length of the posterior neurocranium and (**E**) the trabeculae were not affected. Ethmoid plate in length (**F**) and width (**G**) was reduced by exposure to 25 µM or 50 µM nicotine. Sample sizes are as follows: DMSO n = 8; 12.5 µM nicotine n = 6; 25 µM nicotine n = 7; and 50 µM nicotine n = 8; epl, plate length; epw, plate width; tr, trabecula; n, neurocranium; an, anterior neurocranium; pn, posterior neurocranium. * = *p* < 0.05; ** = *p* < 0.01; *** = *p* < 0.001; **** = *p* < 0.0001. All images were captured at 10× magnification. Scale bar 50 µm.

**Figure 5 toxics-10-00612-f005:**
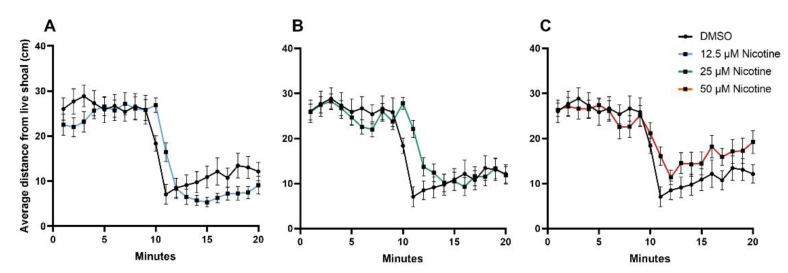
Embryonic nicotine exposure reduces the shoaling response in adult zebrafish. Average distance between the experimental fish and the live shoal, plotted by minute of the 20-min behavioral session. Mean ± SEM is shown. A t-test was used to calculate significance between two groups. (**A**) Comparison between DMSO (n = 22) and 12.5 µM Nicotine (n = 22) groups (*p* = 0.2717). (**B**) Comparison between DMSO and 25 µM Nicotine (n = 30) groups (*p* = 0.4708). (**C**) Comparison between DMSO and 50 µM nicotine (n = 23) groups (*p* = 0.0499).

**Figure 6 toxics-10-00612-f006:**
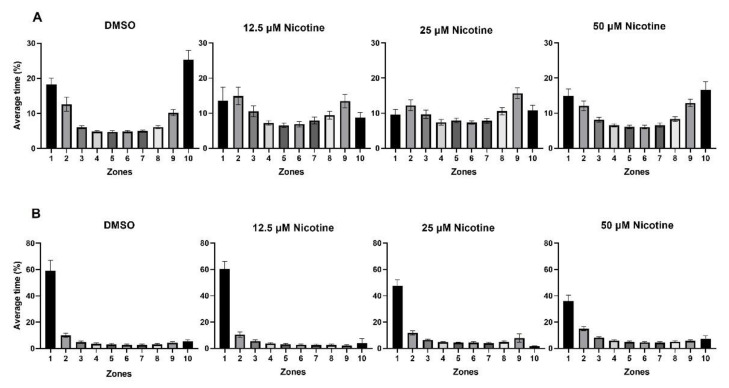
Embryonic nicotine exposure reduces the time fish spend near a live shoal. (**A**) Bars represent the time spent in each of the 10 zones during the first 10 min of habituation and (**B**) the last 10 min of the behavioral session when the stimulus is visible. During habituation, the fish swam back and forth in the tank and spent more time in the zones nearest either end of the tank. During the stimulus, fish from all groups spent more time in zone 1; however, the fish from 50 µM nicotine group spent 23% less time in zone 1 than the control group.

**Figure 7 toxics-10-00612-f007:**
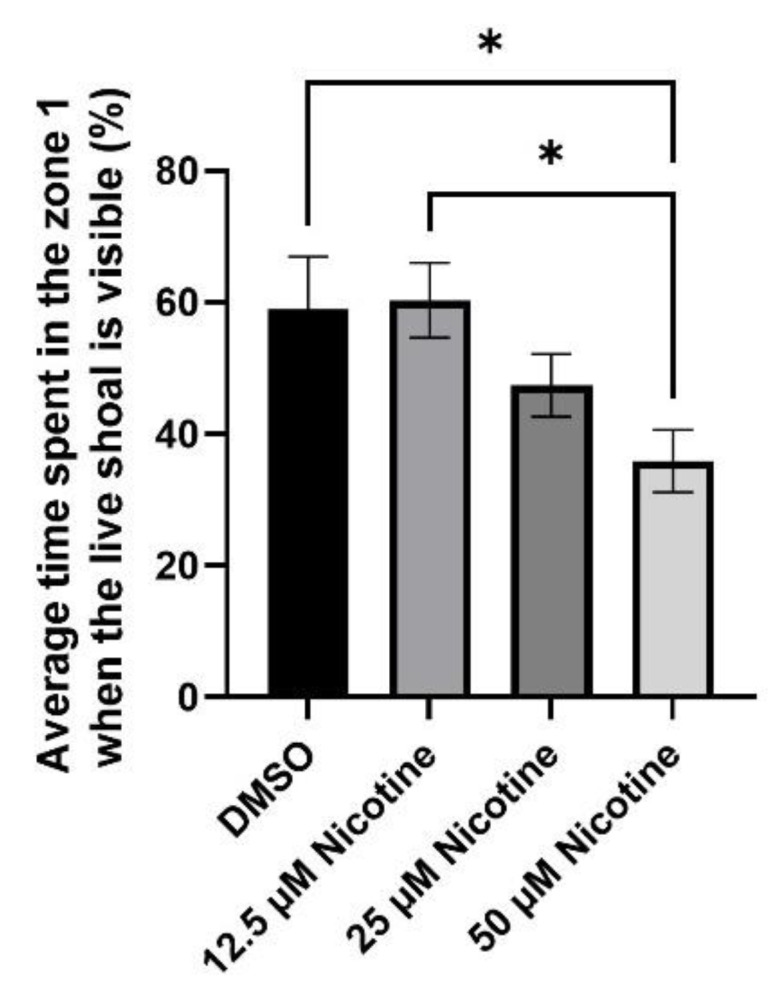
Embryonic exposure to 50 µM nicotine significantly reduces the time spent near a live shoal. Bars represent the average percentage of time spent in the zone closest to the live shoal while the live shoal was visible. * = *p* < 0.05.

**Figure 8 toxics-10-00612-f008:**
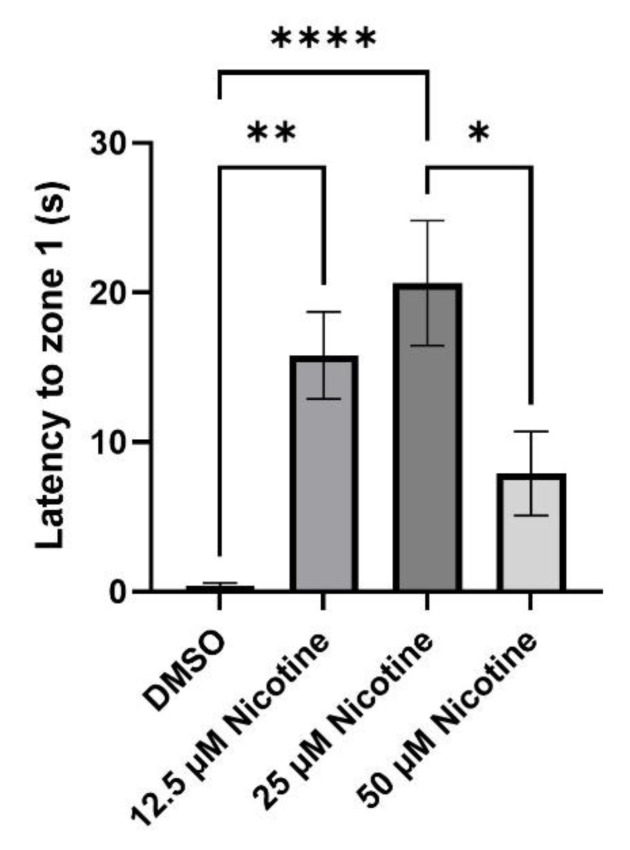
Embryonic nicotine exposure alters the immediate response to a social stimulus in adult zebrafish. Bars represent the time fish take to reach zone 1 once the shoal is visible. While 50 µM nicotine exposure does not change the time to reach 1 one, both the 12.5 µM and 25 µM groups were slower to reach zone 1. * = *p* < 0.05; ** = *p* < 0.01; **** = *p* < 0.0001.

**Figure 9 toxics-10-00612-f009:**
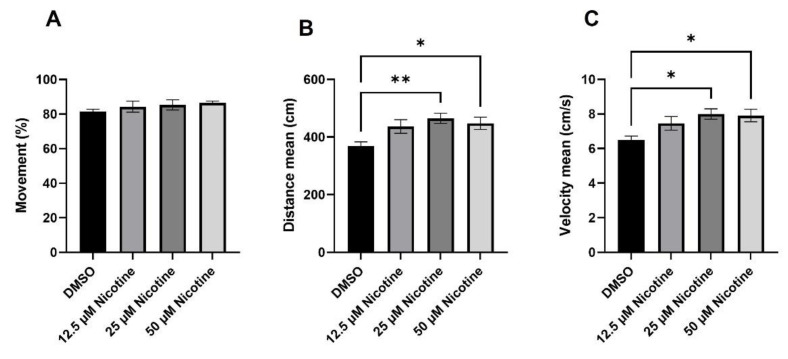
Embryonic nicotine exposure increases locomotion activity in zebrafish adults. Bars represent (**A**) percentage of time the fish is moving, (**B**) the total distance traveled, and (**C**) velocity, during the first 10 min of the behavioral session without stimulus. Embryonic nicotine exposure at 25 µM and 50 µM increases distance traveled and velocity, but does not affect the movement percentage. * = *p* < 0.05; ** = *p* < 0.01.

## Data Availability

Not applicable.

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
