# Peer review of "Embryonic Nicotine Exposure Disrupts Adult Social Behavior and Craniofacial Development in Zebrafish"

_toxics, 2022, doi:10.3390/toxics10100612_

Round 1
Reviewer 1 Report
Review of Borrego-Soto and Eberhart
This paper describes early effects of nicotine exposure on zebrafish larvae cranial development and longitudinal effects on behavior in adult fish. The paper is well written and organized and quite straightforward in its presentation. Not enough longitudinal studies involving behavior and physiology on zebrafish exposed to toxins during early development. The paper represents significant work and should be published. I enjoyed reading it!
I have one concern, which can be easily addressed (I hope). The authors go to great length measuring skeletal length. I was wondering about overall body length of the treated groups. If the animals are smaller in general, it calls in to question the conclusion of specific cranial defects. Can the authors include a sentence that includes overall body length of the different groups? If there are specific craniofacial differences at 4 dpf, a picture of the actual treated fish would also be helpful.
Data on the longitudinal fish was interesting. Survival is an interesting question. It has always struck me as odd that a clutch of siblings from an inbred stock line can vary so greatly in their response to global toxin exposure as embryos. Presumably the survivors are the most resilient “super” fish. Did the authors look at size of adults as a health indicator (skewed by density)? Another confounding issue might be tank density the fish were raised at as a factor in the shoaling. Can the authors mention anything they did for that as a consideration, or as a future parameter to examine?
The one question that I had was that it appeared that the 12.5 µM fish actually seemed to have HIGHER shoaling behavior over the last 7 minutes or so (Figure 5A). Through that same window the 25µM fish appear equal to controls and the 50µM fish actually show an obvious aversion to shoaling. Why choose the 10-minute mark as a cut-off for your analysis? Can you add something to the discussion about this?
Author Response
We thank the reviewers for their time and thoughtful consideration of our manuscript. We have dealt with the reviewer comments as noted below and hope they agree that the manuscript is markedly improved.
Reviewer 1:
1) An overall reduction in size could underly the reduced craniofacial skeleton.
This is an excellent point. We have measured overall body length across treatment groups and have found no differences. These data can be found in the manuscript on lines 222-226 and a figure in supplementary material (S5).
2) Did the authors look at size of adults as a health indicator (skewed by density)? Another confounding issue might be tank density the fish were raised at as a factor in the shoaling. Can the authors mention anything they did for that as a consideration, or as a future parameter to examine?
All the fish in this study were kept in medium-sized tanks, in which the number of adult fish indicated by the manufacturer is 15. For all tanks the density was between 8 and 15 fish. We observed no obvious differences in the size and health of the adult fish in the experimental groups. Our IACUC protocol requires daily health checks and we observed no sickly fish across these groups. We have added these details in lines 78-81.
3). The one question that I had was that it appeared that the 12.5 µM fish actually seemed to have HIGHER shoaling behavior over the last 7 minutes or so (Figure 5A). Through that same window the 25µM fish appear equal to controls and the 50 µM fish actually show an obvious aversion to shoaling. Why choose the 10-minute mark as a cut-off for your analysis? Can you add something to the discussion about this?
We followed a well-established method from Robert Gerlai’s lab., which uses the 10 minute cut off. We note this on lines 117 and 118.
While it appears that there might be a trend for the DMSO group to have less shoaling behavior than the 12.5 µM nicotine group, we found statistical differences after minutes 10 and 11 where the DMSO fish is closer to the stimulus than 12.5 µM fish. This can be found on lines 258-261 and in supplementary material S6.

Reviewer 2 Report
In this manuscript, the authors examined the effects of embryonic nicotine exposure on craniofacial development, as measured in larvae, and on social behavior in adults. They also provide evidence for nicotine metabolism to cotinine in zebrafish. Similar work has previously been reported (see below) although different parameters had been cited. See references below. Overall, the individual experiments in the current study are well documented and a good number of parameters have been determined. The authors are at times a bit bold in their conclusion, especially considering that the work and its interpretation are, at this stage, rather superficial (see specific comments). The authors may wish to perform some of the simple experiments/observations I suggest before the publication of their study although I do not make this a condition for acceptance, provided reasonable disclaimers on the study’s limitations are made.
- It is somewhat disappointing that the authors did not do a simple survey of the literature on nicotine exposure in zebrafish. The authors must mention at least the following two papers in their introduction and discussion: a) Parker and Connaughton Zebrafish 2007;4(1):5doi: 10.1089/zeb.2006.9994. b) Stewart et al., Pharmacol Biochem. Behav. 2015 Dec;139 Pt B:112-20. doi: 10.1016/j.pbb.2015.01.016.
- The authors show that craniofacial development was affected by nicotine exposure, with several, but not all, skeletal elements reduced in size. Despite this non-uniform effects of nicotine on craniofacial development, were the authors able to rule out that the effects that they see are not attributable to an overall developmental delay?. This should be briefly discussed. Have tehy examined older individuals to determine if this effect persists? In the negative, a mention to this effect should be made. This is particularly important considering the authors attribute high mortality in one of their experimental groups to craniofacial defects.
- The authors also argue that neural-crest skeletal elements are particularly sensitive to disruption (lines 209-210). Did the authors examine other neural-crest-derived structures, such as pigmentation?. This should be briefly discussed.
- Although the authors are possibly right in attributing the higher mortality in the 50 micromolar nicotine group to the craniofacial abnormalities they suffered, it doe snot look like feeding behavior was examined in the animals. A mention should be made to this effect.
- Line 306-307: I think the sentence is a bit strong as the measurements are, at this stage, rather primitive. “may cause some hyperactivity” might be in order.
- Line 379: I would prefer “impacts” to “lessens”.
Author Response
We thank the reviewers for their time and thoughtful consideration of our manuscript. We have dealt with the reviewer comments as noted below and hope they agree that the manuscript is markedly improved.
Reviewer 2:
1) The authors must mention at least the following two papers in their introduction and discussion: a) Parker and Connaughton Zebrafish 2007;4(1):5doi: 10.1089/zeb.2006.9994. b) Stewart et al., Pharmacol Biochem. Behav. 2015 Dec;139 Pt B:112-20. doi: 10.1016/j.pbb.2015.01.016.
We thank the reviewer for noting our omission of Parker and Connaughton and have added a discussion in lines 406-407 and 414-420. The Stewart et al. work was not originally cited as it dealt with adult exposures. To be more comprehensive we have added discussion of this work on lines 430-432. We also added other references about nicotine exposures in adult zebrafish and anxiogenic behavioral responses. (Hawkey et al., Subchronic Effects of Plant Alkaloids on Anxiety-like Behavior in Zebrafish. Pharmacology Biochemistry and Behavior 2021, 207, 173223, doi:10.1016/j.pbb.2021.173223. Alzualde et al., Effects of Nicotinic Acetylcholine Receptor-Activating Alkaloids on Anxiety-like Behavior in Zebrafish. J Nat Med 2021, 75, 926–941, doi:10.1007/s11418-021-01544-8).
2) Despite this non-uniform effect of nicotine on craniofacial development, were the authors able to rule out that the effects that they see are not attributable to an overall developmental delay?
To determine if there was an overall developmental delay, we measured the total body length at 4 dpf and found no differences. These data are on lines 222-226 and in supplementary material (S5).
3) The authors also argue that neural-crest skeletal elements are particularly sensitive to disruption (lines 209-210). Did the authors examine other neural-crest-derived structures, such as pigmentation?
We thank the reviewer for this suggestion. As suggested, we quantified melanocytes at 36 hpf in the control group and 50 µM nicotine exposed group. We found a statistically significant reduction in the number of melanocytes in the head of the treated group versus the control group (P = 0.0474; DMSO M = 8.6, SEM ± 0.53; 50 µM Nicotine M = 6.8, SEM ± 0.45). These data can be found in lines 215-220 of the results, 404-407 of the discussion and Supplementary material (S4).
4) Although the authors are possibly right in attributing the higher mortality in the 50 micromolar nicotine group to the craniofacial abnormalities they suffered, it does not look like feeding behavior was examined in the animals. A mention should be made to this effect.
The reviewer is correct. We added this caveat in lines 411-413.
5&6) Line 306-307: I think the sentence is a bit strong as the measurements are, at this stage, rather primitive. “may cause some hyperactivity” might be in order. Line 379: I would prefer “impacts” to “lessens”.
We have made both of these changes. (327 and 468 respectively)

Reviewer 3 Report
Authors reported detailed information on the effects of nicotine to developing zebrafish on craniofacial development and behavioral activity in after growing into an adult. Nicotine is a major ingredient of smoke that can cause severe harmful effects on fetus. Data presented are important especially for nicotine metabolism in developing zebrafish. Unfortunately, however, each data (metabolism, craniofacial effects and behavioral effects) are dependent each other. Authors should add discussion to connect these observations.
1) Authors mentioned the effects of smoking by pregnant mother on neurobehavioral activity. However, effects of nicotine is major in fact in the manuscript. Authors should add some sentences on craniofacial effects. Authors including me cannot understand at first why there are many images of Alcian blue staining before behavioral data.
2) Data on conversion of nicotine to cotinine is beautiful. But, Authors had better connect this data to toxicological endpoints. It was reported that cotinine can affect anxiety-driven zebrafish behavior (doi: 10.1007/s11418-021-01544-8). Authors should add some discussion on the possible effects of cotinine.
3) While Authors mentioned Alizarin red staining in Abstract and Methods, no images are presented in both regular manuscript and supplementary data.
4) In Figure 5 and Figure 6A, effects of 50 µM nicotine seem to be weaker than 12.5 and 25 µM. How can Authors explain these observations?
5) A sentence from L370: Why are CNCC-derived structures particularly sensitive to nicotine? Do Authors speculate that nicotine affect CNCC more selectively? If so, are there any evidence or speculation?
Author Response
We thank the reviewers for their time and thoughtful consideration of our manuscript. We have dealt with the reviewer comments as noted below and hope they agree that the manuscript is markedly improved.
Reviewer 3:
1). Authors should add some sentences on craniofacial effects.
We have expanded our discussion of craniofacial defects associated with smoking during pregnancy in the introduction (lines 47-49) to better guide the reader as suggested.
2) Data on conversion of nicotine to cotinine is beautiful. But, authors had better connect this data to toxicological endpoints. It was reported that cotinine can affect anxiety-driven zebrafish behavior (doi: 10.1007/s11418-021-01544-8). Authors should add some discussion on the possible effects of cotinine.
This is an excellent point and we thank the reviewer for their suggestion. We added a paragraph in lines 360-374 to discuss nicotine-cotinine toxicological effects and we added the reference 10.1007/s11418-021-01544-8 in 361 line.
3) While Authors mentioned Alizarin red staining in Abstract and Methods, no images are presented in both regular manuscript and supplementary data.
We routinely follow Walked and Kimmel protocol for Dual Bone and Cartilage Staining. However, at the time we imaged the zebrafish there is actually very little bone. The opercle structure can be observed in slightly red color in figure 2 in DMSO whole mount and the teeth in DMSO viscerocranium.
4) In Figure 5 and Figure 6A, effects of 50 µM nicotine seem to be weaker than 12.5 and 25 µM. How can Authors explain these observations?
It was surprising to us too that the 50 µM group more closely resembled the DMSO group with regard to shoaling. This effect is difficult to explain. The low survival in this group could, conceivably, have selected for less disrupted individuals. However, this group showed similar effects on swimming velocity and distance. Our discussion of this is on lines 439-451.
5) Why are CNCC-derived structures particularly sensitive to nicotine? Do Authors speculate that nicotine affect CNCC more selectively? If so, are there any evidence or speculation?
Based on this and comments from reviewer 2, we analyzed an additional neural crest-derived cell type, melanocytes, at 36 hpf in the control group and 50 µM nicotine exposed group. We found a statistically reduced number of melanocytes in the head of the treated group versus the control group (P = 0.0474; DMSO M = 8.6, SEM ± 0.53; 50 µM Nicotine M = 6.8, SEM ± 0.45). These data are on lines 216-221 and Supplementary material (S4).

Round 2
Reviewer 2 Report
The authors have adequately addressed the points I raised upon review of the original manuscript.
Reviewer 3 Report
No comments.